# Ultralow thermal conductivity from transverse acoustic phonon suppression in distorted crystalline $\alpha$-MgAgSb

Xiyang Li [1,2,3,4,12], Peng-Fei Liu [5,12], Enyue Zhao[1,4], Zhigang Zhang[2], Tatiana Guidi [6], Manh Duc Le [6], Maxim Avdeev [7], Kazutaka Ikeda[8], Toshiya Otomo [8], Maiko Kofu[9], Kenji Nakajima [9], Jie Chen[5], Lunhua He[1,5], Yang Ren[10], Xun-Li Wang [3], Bao-Tian Wang [5✉], Zhifeng Ren[11✉], Huaizhou Zhao[1✉] & Fangwei Wang [1,2,4✉]

Low thermal conductivity is favorable for preserving the temperature gradient between the two ends of a thermoelectric material, in order to ensure continuous electron current generation. In high-performance thermoelectric materials, there are two main low thermal conductivity mechanisms: the phonon anharmonic in PbTe and SnSe, and phonon scattering resulting from the dynamic disorder in $AgCrSe_2$ and $CuCrSe_2$, which have been successfully revealed by inelastic neutron scattering. Using neutron scattering and ab initio calculations, we report here a mechanism of static local structure distortion combined with phonon-anharmonic-induced ultralow lattice thermal conductivity in $\alpha$-MgAgSb. Since the transverse acoustic phonons are almost fully scattered by the compound's intrinsic distorted rocksalt sublattice, the heat is mainly transported by the longitudinal acoustic phonons. The ultralow thermal conductivity in $\alpha$-MgAgSb is attributed to its atomic dynamics being altered by the structure distortion, which presents a possible microscopic route to enhance the performance of similar thermoelectric materials.

[1] Beijing National Laboratory for Condensed Matter Physics, Institute of Physics, Chinese Academy of Sciences, Beijing 100190, China. [2] Songshan Lake Materials Laboratory, Dongguan 523808, China. [3] Department of Physics, City University of Hong Kong, 83 Tat Chee Avenue, Hong Kong, China. [4] School of Physical Sciences, University of Chinese Academy of Sciences, Beijing 101408, China. [5] Spallation Neutron Source Science Center, Dongguan 523803, China. [6] ISIS facility, Rutherford Appleton Laboratory, Chilton, Didcot OX11 0QX Oxfordshire, UK. [7] Australian Nuclear Science and Technology Organisation, Lucas Heights, NSW 2234, Australia. [8] Institute of Materials Structure Science, High Energy Accelerator Research Organization (KEK), Tsukuba, Ibaraki 305-0801, Japan. [9] Japan Proton Accelerator Research Complex, Japan Atomic Energy Agency, Tokai, Ibaraki 319-1195, Japan. [10] X-ray Science Division, Argonne National Laboratory, Argonne, IL 60439, USA. [11] Department of Physics and TcSUH, University of Houston, Houston, Texas 77204, USA. [12] These authors contributed equally: Xiyang Li, Peng-Fei Liu. ✉email: wangbt@ihep.ac.cn; zren@uh.edu; hzhao@iphy.ac.cn; fwwang@iphy.ac.cn

Transverse acoustic phonons are believed to compete with structure disorder, such as in superionic crystals[1,2], glasses[3], liquids[4], and model crystal-like aperiodic solids[5]. The main heat carriers in thermoelectric materials are acoustic phonons[1,6]. Thermoelectric materials, which can be used to directly convert thermal energy and electrical energy, have attracted much attention for meeting current and future energy demands[6–9]. The thermoelectric conversion efficiency is governed by the material's figure of merit[7], $ZT = [S^2\sigma/(\kappa_{lat} + \kappa_{ele})]T$, where $S$, $\sigma$, $\kappa_{lat}$, $\kappa_{ele}$, and $T$ are the Seebeck coefficient, electronic conductivity, lattice thermal conductivity, electronic thermal conductivity, and absolute temperature, respectively. Low thermal conductivity is one of the most vital properties of high-performance thermoelectric materials[1,7,10,11]. Besides band engineering, which can enhance a material's electrical transport properties, many manipulations have been studied with the aim of modulating the $\kappa_{lat}$, including nano-crystallization[12–14], crystal defects[15,16], structure disorder[1,17], rattling guest-filling[18,19] among other techniques, which are beneficial for increasing phonon scattering[1,7,13,20]. On the other hand, materials with intrinsically low $\kappa_{lat}$, such as PbTe[10,21], SnSe[22,23], BiSe[24], skutterudites[18], $Bi_2Te_3$[25], $MCrSe_2$ ($M$ = Ag or Cu)[1,2], and MgAgSb[15], are of great interest[11]. Thus, it is both scientifically and technologically significant to study the structure and atomic dynamics of high-performance thermoelectric materials.

The high-performance MgAgSb-based thermoelectric materials have great potential as candidates for near-room temperature (RT) thermoelectric generators[15,26]. Their $ZT$ values reach ~0.9 at 300 K, and a maximum of 1.4 at 453 K, which fills the materials gap between low-temperature $Bi_2Te_3$ alloys and the middle-temperature PbTe systems in the $ZT$ spectrum[10,15,25,27]. A record high thermoelectric conversion efficiency of 8.5% with a single thermoelectric leg operating at between 293 and 518 K has been achieved[28]. Recently, an improved $ZT$ of 2.0 and conversion efficiency of 12.6% in Zr- and Pd-doped MgAgSb $p$-type materials were predicted theoretically[29]. The $\kappa_{lat}$ of these materials is 0.4 ~0.5 Wm$^{-1}$K$^{-1}$, comparable with that of the ultralow $\kappa_{lat}$ in SnSe originated from the strong lattice anharmonicity[22,27]. This material has three types of structures at different temperatures, the half-Heusler structure $\gamma$-MgAgSb at high temperatures >633 K, the $Cu_2Sb$-related structure $\beta$-MgAgSb at intermediate temperatures between 633 and 563 K, and the tetragonal structure $\alpha$-MgAgSb at low temperatures between 563 and 303 K[30]. A distorted Mg–Sb rocksalt-type sublattice can be formed in $\alpha$-MgAgSb[30]. Study of the detailed structure and dynamics of the MgAgSb-based materials is vital to understand the origin of their high-performance thermoelectric properties, with emphasis on their low $\kappa_{lat}$[1,10,12,27,31].

Thus far, in addition to transport property measurements, experimental characterizations of the MgAgSb-based materials have been mainly based on X-ray diffraction and electron microscopy, which have provided microscopic insight into their crystalline structures[15,26,30,32]. In contrast, numerous theoretical characterizations have focused on their local crystal structure, electronic band structure, chemical bonding, and atomic dynamics, etc.[26,27,29,33]. Additionally, to date, there have only been theoretical calculations of the phonon modes for the MgAgSb family[29,34] without any experimental verification. A detailed atomistic understanding of the ultralow $\kappa_{lat}$ in $\alpha$-MgAgSb has remained elusive due to the lack of such phonon measurements. Fortunately, new chopper spectrometers at state-of-the-art high-flux neutron sources coupled with advances in high-resolution neutron instruments have enabled high-precision measurements of the dynamic structure factor, $S(\mathbf{Q}, E)$, which contains information on the atomic dynamics[1,2,10,23].

Here, we use neutron scattering measurements together with systematic ab initio simulations to study the crystalline structure and the atomic dynamics of two $\alpha$-MgAgSb-based materials, MgAg$_{0.97}$Sb$_{0.99}$ and MgAg$_{0.965}$Ni$_{0.005}$Sb$_{0.99}$ (the sample of Ni $p$-type substitution for Ag has a higher anomalous electrical resistivity[15]), the $\alpha$-phase of which exhibits the highest thermoelectric performance[32]. We find that their ultralow $\kappa_{lat}$ is induced both by static local structure distortion suppression of the transverse acoustic phonons and the phonon anharmonicity.

## Results

**Crystallographic structure properties**. The evolution of the crystallographic structures of MgAg$_{0.97}$Sb$_{0.99}$ and MgAg$_{0.965}$-Ni$_{0.005}$Sb$_{0.99}$ as a function of temperature was investigated by neutron diffraction. Both compounds maintain the $\alpha$-MgAgSb structure (Fig. 1a) over a wide temperature range from 20 to 500 K (Fig. 1b, c and Supplementary Fig. 1a, b). The structure features a 24-atom trigonal primitive unit cell with a large distorted rocksalt sublattice, where the Ag atoms fill half of the Mg–Sb distorted cubes. This structure favors the low thermal conductivity paradigm of crystals with complex unit cells[17]. The distorted structure has a significant phonon scattering effect and the complex primitive unit cell has a large ratio of optical phonon branches (69/72) that significantly reduces $\kappa_{lat}$[17], since acoustic phonons are the main contributor to $\kappa_{lat}$[1,6,20].

The temperature-dependent neutron diffraction data (Fig. 1c and Supplementary Fig. 1b) were analyzed by the method of Rietveld refinement, and the lattice thermal dilatation shows anisotropic features as revealed by the increasing value of $c/a$ with increasing temperature (Supplementary Fig. 1f). The lattice thermal expansion has an inflection point at ~200 K (Supplementary Fig. 1c, d), which is consistent with the previously measured thermal conductivity and $ZT$ data[27]. This indicates that an additional phonon-scattering mechanism related to the structure evolution may appear at high temperatures.

**Local structure distortion**. Here, we study the local structure of MgAg$_{0.965}$Ni$_{0.005}$Sb$_{0.99}$ using neutron total scattering. By the pair distribution function (PDF) analysis and fitting of $G(r)$, which is a Fourier transform result of the static structure factor, $S(Q)$ (see Methods section)[1,20,35], we directly reveal the distortion of the Mg–Sb rocksalt sublattice in real space by the shoulder peak at ~3.4 Å and the small peak at ~5.8 Å (Fig. 2a). The refined Mg–Sb bond distances are in the range from 2.86(2) Å to a large value of 3.90(2) Å, which is larger than the radius summation of Mg and Sb atoms (Mg: 1.60 Å, Sb: 1.44 Å; Fig. 2b). It reveals a strong distortion and weak bonding nature of Mg–Sb bonds, which indicates that novel atomic dynamics may exist[23].

**Transverse acoustic phonon suppression from local structure distortion**. Consequently, we investigated the atomic dynamic properties of these materials by inelastic neutron scattering (INS). Figure 3a shows a representative data set of the Bose-factor-calibrated dynamic structure factor, $B(\mathbf{Q}, E) = \left[1 - e^{-\frac{E}{k_B T}}\right] S(\mathbf{Q}, E)$, for the MgAg$_{0.965}$Ni$_{0.005}$Sb$_{0.99}$ compound, obtained with incident neutron energy of $E_i = 15.16$ meV at 300 K (see Methods section). The $S(Q)$ data for the same sample measured at 300 K were also plotted at the bottom of Fig. 3a for comparison. The intensity of the sharpest peak at $Q$ ~2.76 Å$^{-1}$ for $S(Q)$ is about four times as strong as the second sharpest one at $Q$ ~2.17 Å$^{-1}$. This can be ascribed to the particular distorted crystal structure, which has many equivalent Brillouin zone centers in different reciprocal space directions folded at this sharpest peak $Q$ position. We consider these centers to be a quasi-Brillouin-zone (QBZ) center. Thus, a

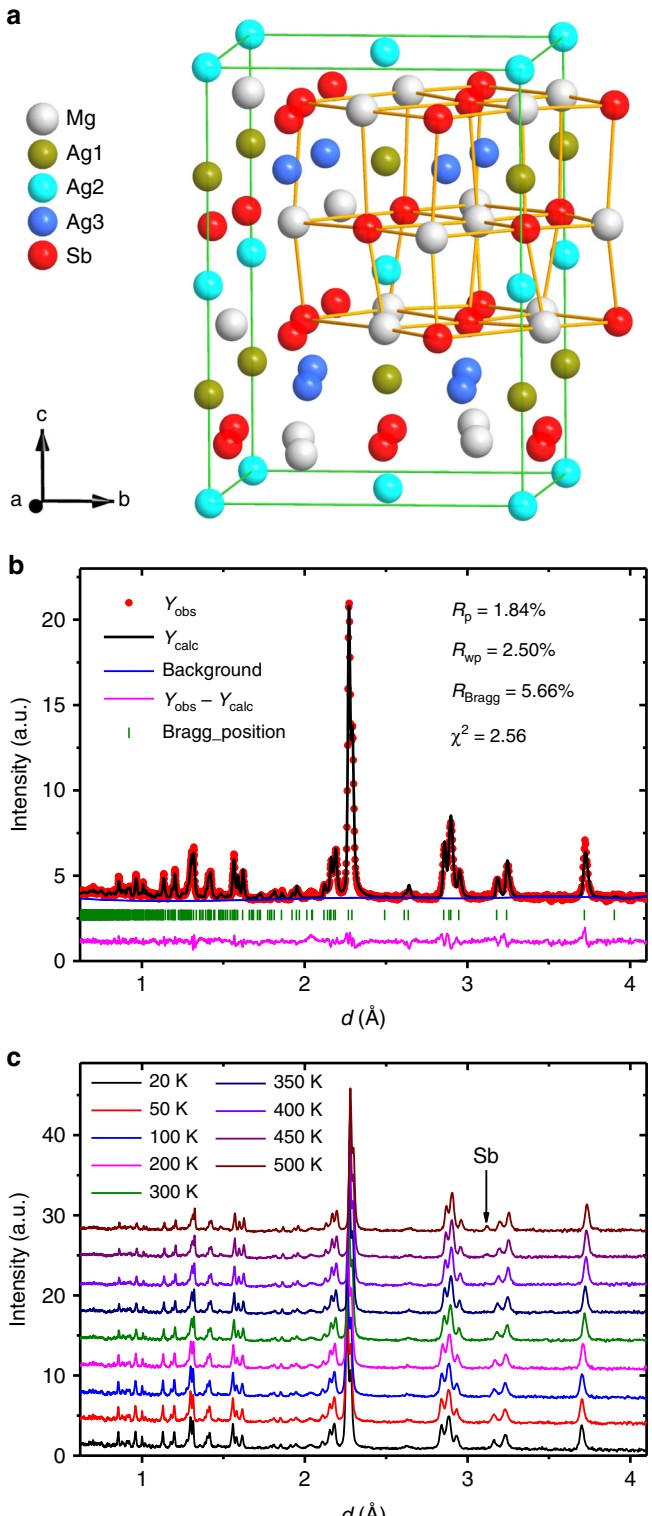

**Fig. 1 Neutron diffraction measurements demonstrate that MgAg$_{0.965}$Ni$_{0.005}$Sb$_{0.99}$ maintains the $\alpha$-MgAgSb tetragonal structure from 20 to 500 K. a** Crystalline structure of $\alpha$-MgAgSb with a complex distorted Mg–Sb rocksalt sublattice, where half of the Mg–Sb distorted cubes are filled with Ag atoms. **b** Rietveld refinement of MgAg$_{0.965}$Ni$_{0.005}$Sb$_{0.99}$ neutron diffraction data measured at 300 K. **c** Temperature-dependent MgAg$_{0.965}$Ni$_{0.005}$Sb$_{0.99}$ neutron diffraction data. These data reveal that MgAg$_{0.965}$Ni$_{0.005}$Sb$_{0.99}$ maintains the same $\alpha$-MgAgSb tetragonal structure with a space group of $I−4c2$ (no.120) over a temperature range from 20 to 500 K. The 450 and 500 K data show the appearance of a small amount of Sb precipitate; a.u., arbitrary units.

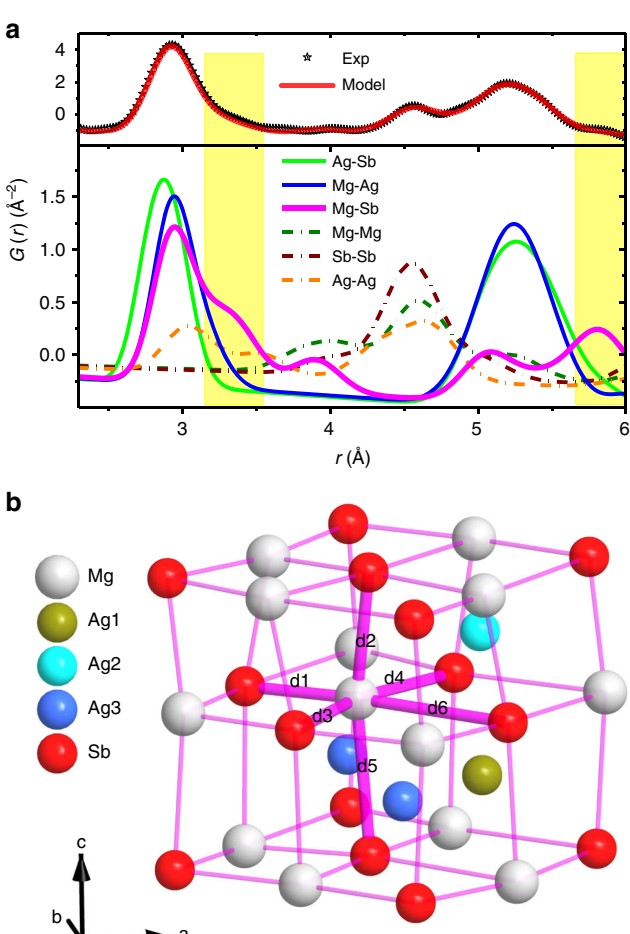

**Fig. 2 Partial PDF directly reveals the static local structure distortion of the Mg-Sb rocksalt sublattice. a** Real space $G(r)$ fitting demonstrates that the first sharp peak at $r \sim 3.0$ Å mainly results from bonds between distinct-atoms, while the shoulder peak at $r \sim 4.5$ Å mainly results from bonds between atoms of the same elements. The distortion of the Mg–Sb rocksalt results in the split of Mg–Sb nearest-neighbor bonds. Here, this PDF data directly indicates this distortion by the peaks at $r \sim 3.3$ and $\sim 5.8$ Å (yellow-tinted regions), as a result of the distinguishable contribution of Mg–Sb bonds to these peaks. The data are measured at 300 K. **b** Distorted structure of the Ag atom-filled Mg–Sb rocksalt sublattice. The refined nearest Mg–Ag bond distances d1–d6 are between 2.86(2) and 3.90(2) Å.

novel double-mushroom scattering pattern arising from the QBZ center is observed in Fig. 3a. To examine the acoustic nature of the low-energy vibration modes, we using the resonant ultrasound spectrometer measured sound velocities, with $V_T = 1102$ m/s and $V_L = 3708$ m/s[27], calculated the dispersions that are the magenta and green solid lines arising from the QBZ center in Fig. 3a for the transverse and longitudinal acoustic modes, respectively. The $E$-cut data at 2.0, 2.5, and 3.0 meV are shown in Fig. 3d. The green and magenta arrows indicate the longitudinal and transverse phonon peak positions, respectively, which were calculated using the sound

velocities. The ab initio simulation result, using the VASP and OCLIMAX programs (see Methods section), is shown in Fig. 3b. By linking together the INS experimental results, the simulation results (Fig. 3b and Supplementary Fig. 2), and the calculations by the sound velocities, and applying them to the observed

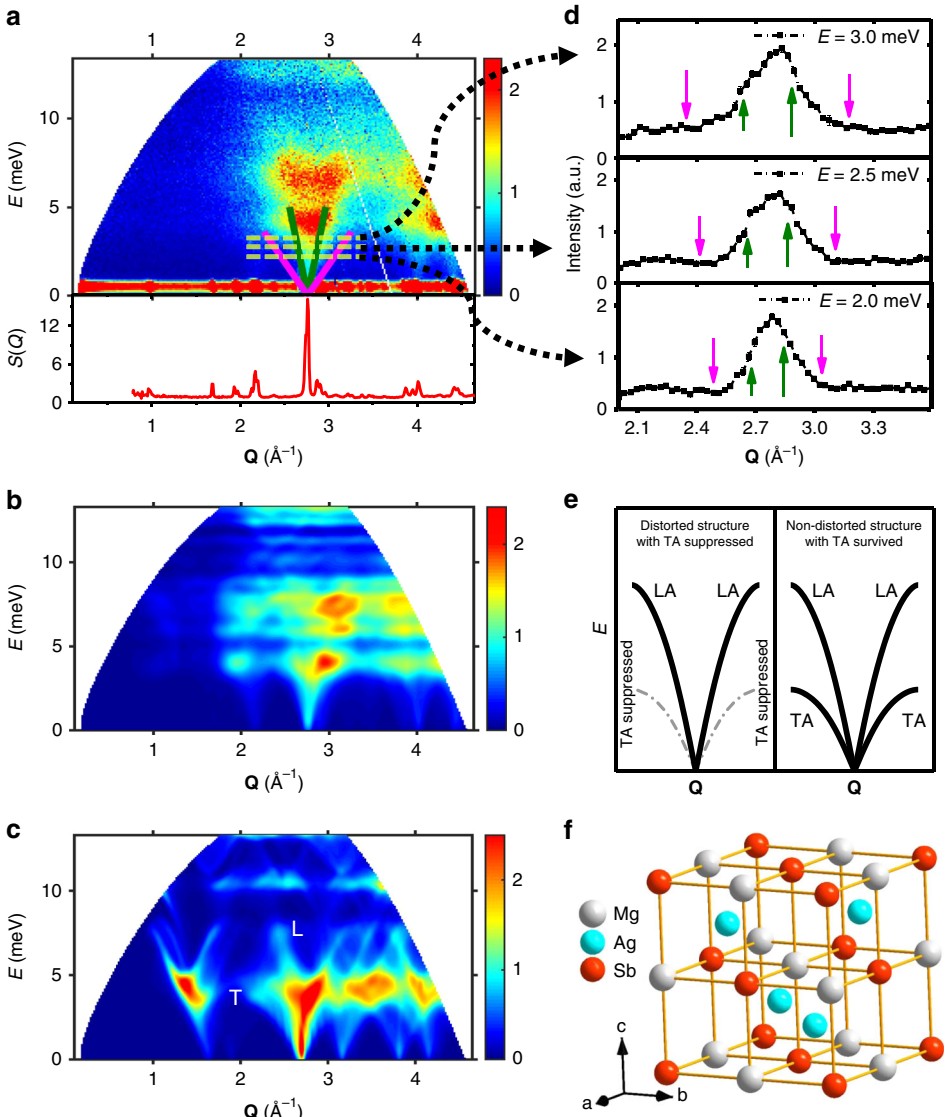

**Fig. 3 Transverse acoustic phonon suppression by the static local structure distortion. a** Bose-factor-calibrated dynamic structure factor, $B(\mathbf{Q}, E)$, measured with INS (top) and static structure factor, $S(Q)$, measured with neutron total scattering (bottom) at 300 K in $MgAg_{0.965}Ni_{0.005}Sb_{0.99}$. The magenta and green lines are calculated dispersions based on transverse and longitudinal sound velocities, respectively, which were measured by the resonant ultrasound spectrometer method. **b** The corresponding neutron-weighted ab initio calculated $B(\mathbf{Q}, E)$ pattern using the OCLIMAX program for $\alpha$-MgAgSb. **c** The corresponding neutron-weighted ab initio calculated $B(\mathbf{Q}, E)$ pattern based on an adjusted symmetric Ag atom-filled Mg–Sb rocksalt structure. **d** $E$-cut data at 2.0, 2.5, and 3.0 meV. The magenta and green arrows indicate the transverse and the longitudinal phonon peak positions, respectively, which are calculated by the sound velocities. Error bars are propagated from counting statistics on measured spectra; a.u., arbitrary units. **e** A schematic shows transverse acoustic phonon suppression by the static local structure distortion. **f** The adjusted symmetric Ag atom-filled Mg–Sb rocksalt structure used in **c** calculation. By comparing **b** with **c**, the transverse acoustic phonons are observed to survive in the symmetric structure while they disappear in the distorted structure **e**. These results demonstrate that the transverse acoustic phonons are mostly suppressed by the distorted structure in this material. For the purpose of comparison, the color bars in **a**, **b**, and **c** are plotted in relative intensities with arbitrary units.

double-mushroom scattering pattern (Fig. 3a), it was determined that the lower branches at $E \sim 4.5$ meV are mainly the longitudinal acoustic phonon modes, whereas the upper branches at $E \sim 7.0$ meV are mainly the low-energy optical phonon modes (Figs. 3a, 4b). Most importantly, these results illustrate that the transverse acoustic phonons are fundamentally suppressed in this material.

To study the origin of the suppression of the transverse acoustic phonons, we adjusted the distorted Mg–Sb sublattice to a highly symmetric Mg–Sb rocksalt structure by making all of the Mg–Sb bonds equivalent (Fig. 3f). We then calculated the phonon spectrum of this non-distorted structure (Supplementary Fig. 3) and the corresponding scattering pattern (Fig. 3c), using the same method and instrument parameters as in Fig. 3b. Here, the

longitudinal and transverse acoustic branches marked as $L$ and $T$ in Fig. 3c, respectively, arising from the QBZ center, i.e., $Q \sim 2.76$ Å$^{-1}$, can be clearly seen. Magnifications of areas of $B(\mathbf{Q}, E)$ at the $Q$ from 2.2 to 3.3 Å$^{-1}$ and $E$ from 0 to 10 meV region in Fig. 3a–c are shown in Supplementary Fig. 4, making their differences clearly visible. By comparing these high-symmetry structure results with those of the distorted structure, the suppression of the transverse acoustic phonons by the static local structure distortion in $\alpha$-MgAgSb is clear (Fig. 3e). We stress that this suppression results in the ultralow $\kappa_{lat}$ of $\alpha$-MgAgSb, which is different from the case of superionic conductors $AgCrSe_2$[1] and $CuCrSe_2$[2], in which the dynamic disorder with crystal structure transition suppresses the transverse acoustic phonons.

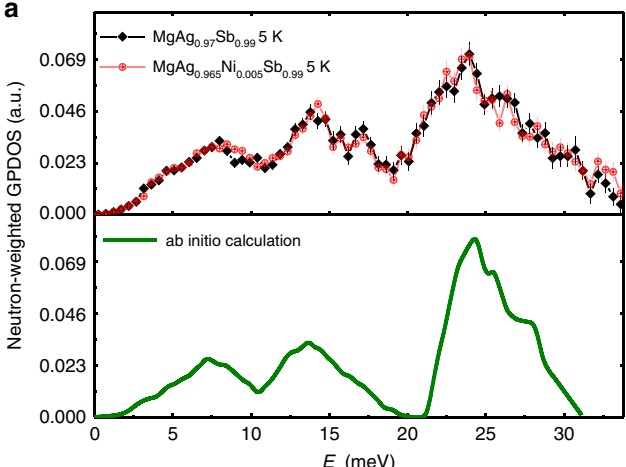

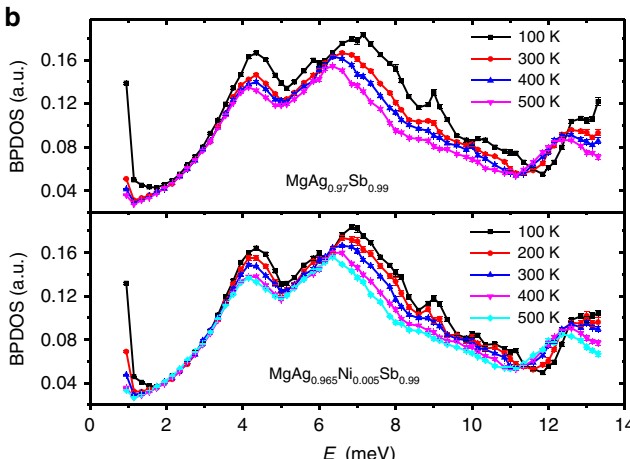

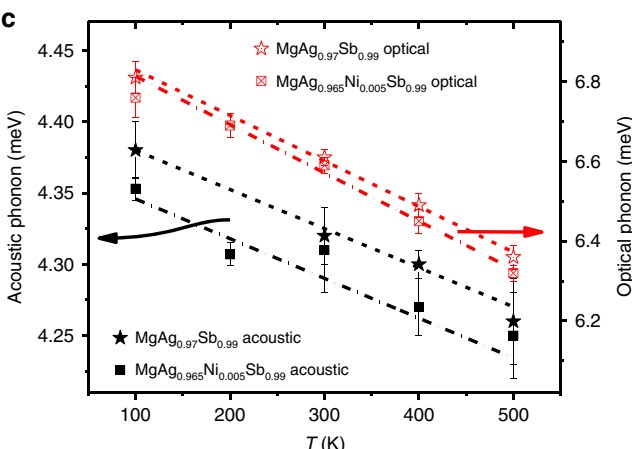

**Fig. 4 Atomic dynamic properties of MgAgSb-based thermoelectric materials. a** Neutron-weighted GPDOS measured by INS in MgAg$_{0.97}$Sb$_{0.99}$ and MgAg$_{0.965}$Ni$_{0.005}$Sb$_{0.99}$ at 5 K and the ab initio calculations for the $\alpha$-MgAgSb tetragonal phase at 0 K. **b** Neutron-weighted low-energy BPDOS measured by high-resolution INS in MgAg$_{0.97}$Sb$_{0.99}$ and MgAg$_{0.965}$Ni$_{0.005}$Sb$_{0.99}$ at selected temperatures. Here, the low-energy peak at ~4.5 meV is mainly due to acoustic phonon modes and the peak at ~7.0 meV is mainly due to low-frequency optical phonon modes. **c** Temperature dependence of the peak positions of acoustic phonons and low-frequency optical phonons, which are fitted from INS data shown in **b**, showing gradual softening. The phonon modes of the Ni-doped sample are much softer than those of the parent sample. The lines are the linear fitting results. The values of the phonon softening ratio (the slope of the fitting lines) are shown in Supplementary Table 3. Error bars in **a** and **b** are propagated from counting statistics on measured spectra and error bars in **c** are result from the statistical uncertainties in fitting the phonon peaks; a. u., arbitrary units.

**Phonon softening**. Figure 4 shows the atomic dynamic properties in the MgAgSb-based thermoelectric materials, which are measured by INS and calculated by ab initio calculations (see Methods section). Here, we demonstrate that the values of the generalized phonon density of states (GPDOS) of both MgAg$_{0.97}$Sb$_{0.99}$ and MgAg$_{0.965}$Ni$_{0.005}$Sb$_{0.99}$ measured at 5 K by INS are in good overall agreement with the simulation results of $\alpha$-MgAgSb, not only in terms of the total shape but also the energy of the main features, including the three main optical phonon peaks at ~8, 14, and 24 meV and the acoustic phonon shoulder peak at ~4.5 meV (Fig. 4a).

Since the $ZT$ value of both MgAg$_{0.97}$Sb$_{0.99}$ and MgAg$_{0.965}$-Ni$_{0.005}$Sb$_{0.99}$ increases with increasing temperature before

reaching a maximum at ~450 K with a plateau from 450 to 550 K[15,27], we performed temperature-dependent high-resolution INS measurements to further study the properties of the low-frequency phonons, especially the acoustic phonons. Figure 4b shows the Bose-factor-calibrated phonon density of states (BPDOS) as a function of temperature (see Methods section). Here, the energy gap between the low-energy optical phonons and the acoustic phonons at ~5 meV is mapped much more clearly, benefitting from the higher resolution. More interestingly, a temperature-induced phonon softening is shown by these data. The peak positions of the acoustic phonon modes and the low-energy optical phonon modes fitted by the Gaussian function are plotted in Fig. 4c. The corresponding ab initio calculation results using the temperature-dependent lattice parameters are shown in Supplementary Fig. 5. As the temperature-induced phonon softening mainly arises from the lattice expansion and anharmonicity[2], the anharmonic nature of $\alpha$-MgAgSb can be verified by comparing the INS and simulation results. We find that the softening ratio of the INS-measured BPDOS as a function of temperature is about two times as strong as that of the corresponding simulation (Fig. 4c, Supplementary Fig. 5, Supplementary Table 3), which reveals its anharmonic nature.

**Low thermal conductivity mechanism**. To further verify the low thermal conductivity mechanism, we computed the intrinsic anharmonic effects of $\alpha$-MgAgSb and the high-symmetry structure MgAgSb (the $\gamma$-MgAgSb shown in Fig. 3f) from first principles using ShengBTE[36] and Phonopy[37]. Figure 5a shows the temperature-dependent $\kappa_{lat}$ of $\alpha$-MgAgSb demonstrating the overall agreement between the calculation results of MgAgSb and the experimental results of MgAg$_{0.97}$Sb$_{0.99}$[15,38]. The calculated RT $\kappa_{lat}$ of $\alpha$-MgAgSb is 0.54 Wm$^{-1}$ K$^{-1}$, which is comparable with the experimental value of MgAg$_{0.97}$Sb$_{0.99}$ (~0.6 Wm$^{-1}$ K$^{-1}$)[15]. As clearly indicated in Fig. 5b, c, the phonon group velocities ($v$) of $\alpha$-MgAgSb are lower than those of $\gamma$-MgAgSb, while its phonon lifetimes ($\tau$) are larger than those of $\gamma$-MgAgSb, especially for the acoustical phonons <10 meV in $\gamma$-MgAgSb. The three-phonon process is easier to occur in $\gamma$-MgAgSb compared to $\alpha$-MgAgSb demonstrated by the calculated three-phonon scattering phase space shown in Supplementary Fig. 6, and accordingly enhance the phonon anharmonicity of $\gamma$-MgAgSb. The total Grüneisen parameter ($\gamma_{total}$) obtained as a weighted sum of the mode contributions at 300 K are 1.51 and 3.03 for $\alpha$-MgAgSb and $\gamma$-MgAgSb, respectively (Fig. 5d). Generally, large $\gamma_{total}$ corresponds to large phonon anharmonicity and low $\kappa_{lat}$, for instance the $\gamma_{total}$ are 1.45 for PbTe[39], 2.83 for SnSe[39], 3.5 for AgSbSe$_2$[40], and 3.9 for

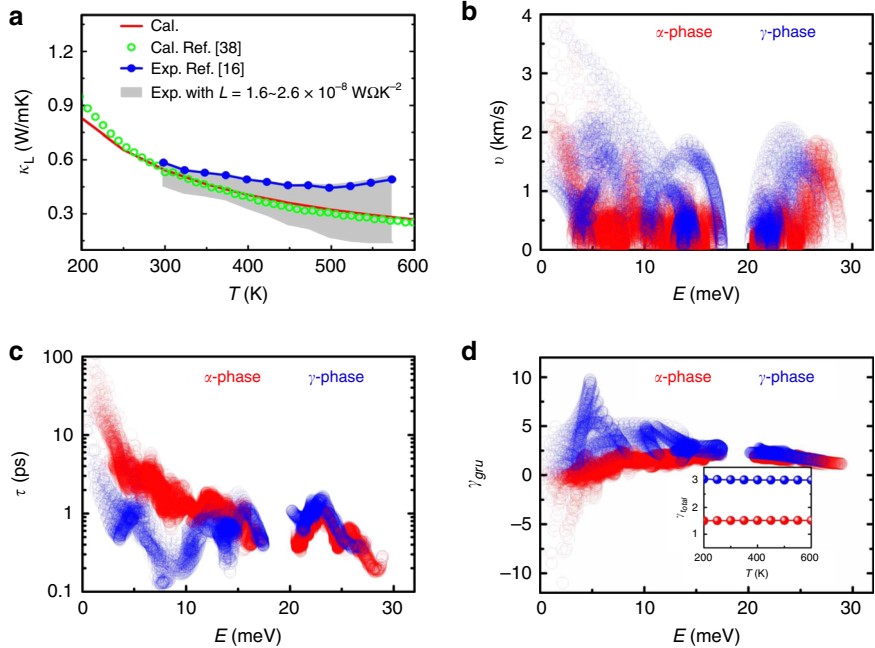

**Fig. 5 Phonon transport properties in $\alpha$-MgAgSb and in the high-symmetry structure MgAgSb (i.e., the $\gamma$-MgAgSb shown in Fig. 3f). a** Experimental and theoretical temperature-dependent thermal conductivity of the $\alpha$-MgAgSb phase. The literature values from ref. [38] (Cal.) and ref. [15] (Exp.) are also plotted for comparison. According to $\kappa_{tot} = \kappa_{lat} + \kappa_{ele} = \kappa_{lat} + L\sigma T$, the lattice thermal conductivity ($\kappa_{lat}$) can be obtained by subtracting $\kappa_{ele}$ from the $\kappa_{tot}$. The shadow regions are the experimental thermal conductivity of MgAg$_{0.97}$Sb$_{0.99}$ with the Lorenz number[7,15] being from 1.6 to $2.6 \times 10^{-8}$ W$\Omega$K$^{-2}$. Here, we only deal with the pure $\alpha$-MgAgSb crystal and consider the phonon–phonon coupling to stimulate the phonon transport properties. Our calculated values are in excellent agreement with the previous reports and our measured values[15,38]. **b** Calculated phonon group velocities $v$, **c** phonon relaxation time $\tau$, and **d** Grüneisen parameters $\gamma_{gru}$ for $\alpha$- and $\gamma$-MgAgSb at 300 K. The inset in **d** shows total $\gamma_{total}$ obtained as a weighted sum of the mode contributions, as a function of temperature for $\alpha$- and $\gamma$-MgAgSb. The $\gamma_{gru}$ of high values accumulate in the vicinity of 5 meV corresponding to the transverse acoustic phonon modes for $\gamma$-MgAgSb.

CsAg$_5$Te$_3$[41], corresponding to measured RT $\kappa_{lat}$ of 2.4, 0.62, 0.48, and 0.2 Wm$^{-1}$ K$^{-1}$, respectively. Such a $\gamma_{total}$ ~1.51 of $\alpha$-MgAgSb should, therefore, give a much larger $\kappa_{lat}$ than that in anomalously high anharmonicity $\gamma$-MgAgSb ($\gamma_{total}$ ~3.03), although its $v$ are slightly lower than those of $\gamma$-MgAgSb. However, our calculations indicate that the $\kappa_{lat}$ of $\alpha$-MgAgSb and $\gamma$-MgAgSb are nearly equal over 300 K with value differences <0.06 Wm$^{-1}$ K$^{-1}$ (Fig. 5a, Supplementary Fig. 7). This, from a side, confirms our hypothesis that the fully scattered transverse acoustic phonons by the static local structure distortion greatly reduce $\kappa_{lat}$ of the weak-anharmonicity $\alpha$-MgAgSb and make it comparable with giant anharmonic materials[23].

## Discussion

Based upon our ab initio simulation results, we determined that the low-energy phonon modes are mainly due to the heavy Ag atom-related vibrations (Supplementary Fig. 8). Interestingly, by Ni doping at the Ag site, the phonon modes of the Ni-doped sample are much softer than those of their parent sample, although the lattice parameters of the Ni-doped sample are slightly smaller (Supplementary Fig. 1). This anomalous phonon softening is consistent with the corresponding anomalous higher electrical resistivity and lower thermal conductivity in the Ni-doped sample[15]. Similar anomalies have been observed in $n$-type Te-doped Mg$_3$Sb$_2$-based materials that are induced by the variations of Hall carrier concentration and mobility[16]. This result reveals the microscopic mechanism of the Ni-doping induced decrease in thermal conductivity and enhancement of thermoelectric performance[15].

In summary, the microscopic origin of the ultralow thermal conductivity in $\alpha$-MgAgSb has been unveiled. Since the transverse acoustic phonons are mostly scattered by the intrinsic Mg–Sb distorted rocksalt sublattice structure, the longitudinal acoustic phonons are consequently responsible for the $\kappa_{lat}$ leading to the ultralow values, and the anharmonic nature of the atomic dynamics is further revealed by the phonon softening. The scenario of ultralow $\kappa_{lat}$ induced by static local structure distortion is of fundamental significance, since it potentially provides a general route to suppress transverse acoustic phonons, and thus the $\kappa_{lat}$ of solids, especially rocksalt-related thermoelectric materials.

## Methods

**Sample synthesis**. The two-step process combining ball milling with hot pressing was used to synthesize the materials MgAg$_{0.97}$Sb$_{0.99}$ and MgAg$_{0.965}$Ni$_{0.005}$Sb$_{0.99}$ as reported elsewhere[15].

**Hall measurements**. The Hall coefficient measurement was performed on Accent HL5500 Hall System for MgAg$_{0.97}$Sb$_{0.99}$ and MgAg$_{0.965}$Ni$_{0.005}$Sb$_{0.99}$ at RT[27]. The resistivity is measured using a four-point method with a sample dimension of 3 × 3 × 0.5 mm$^3$. Hall mobility was determined by $\mu_H = |R_H|/\rho$. The charge-carrier concentration was determined by $n_H = 1/(e/|R_H|)$ based on the one-band model.

**Synchrotron X-ray and neutron diffraction measurements**. Synchrotron X-ray diffraction measurement was carried out at the beamline 11-ID-C at RT at the Advanced Photon Source at Argonne National Laboratory in the U.S. High-energy X-rays with a wavelength of 0.117418 Å and beam size of 0.5 mm × 0.5 mm were used in the transmission geometry for MgAg$_{0.97}$Sb$_{0.99}$ sample data collection.

Neutron powder diffraction measurements were performed using the high-resolution powder diffractometer ECHIDNA at the ANSTO in Australia. A Ge (335) monochromator was used to produce a monochromatic neutron beam of wavelength 1.6215(1) Å. The neutron powder diffraction data were collected at 3 and 300 K for the MgAg$_{0.97}$Sb$_{0.99}$ sample. The FullProf program was used for the Rietveld refinement of both synchrotron X-ray and neutron diffraction data for the crystal structures of the compound. The 150, 250 K, and RT neutron powder diffraction patterns were obtained using the general-purpose powder diffractometer (GPPD)[42] (90° bank) at the China spallation neutron source (CSNS) in China for MgAg$_{0.97}$Sb$_{0.99}$

and MgAg$_{0.965}$Ni$_{0.005}$Sb$_{0.99}$ samples, and the Rietveld refinement is performed via the Z-Rietveld program. The refined results are shown in Supplementary Fig. 1. Another temperature-dependent neutron diffraction measurements were carried out at the high-intensity total diffractometer beamline BL21 NOVA at the J-PARC in Japan at 20, 50, 100, 200, 300, 350, 400, 450, and 500 K for both MgAg$_{0.97}$Sb$_{0.99}$ and MgAg$_{0.965}$Ni$_{0.005}$Sb$_{0.99}$ samples, and the Rietveld refinement is performed by the Z-Rietveld program. The refined parameters are shown in Supplementary Tables 1 and 2. An empty vanadium cell, a standard vanadium rod, and the empty background were also measured at the same temperatures for PDF analysis[35]. Corrections were made for the background, attenuation factor, number of the incident neutron, the solid angle of detectors, multiple scattering, and incoherent scattering cross sections and the intensity was normalized to determine the static structure factor $S(Q)$. The reduced PDF data were calculated by the Fourier transform of $S(Q)$, $G(r) = \frac{2}{\pi} \int_0^\infty Q[S(Q) - 1]sin(Qr)dQ$, with a cutoff ($Q_{max}$) of 30 Å$^{-1}$. The same parameters were applied to the data analysis for both samples. The real-space refinement of experimental $G(r)$ was performed by PDFgui program[43]. In the refinement, the positions of all atoms in unit cell are written and refined. The symmetry constrains are generated by the symmetry of the space group. The values of instrument resolution dampening factor $Q_{damp}$ and resolution peak broadening factor $Q_{broad}$ are determined from standard Si powder data and are fixed in the refinement.

Here, a small impurity-phase peak appears in the 450 and 500 K neutron powder diffraction data (Fig. 1c, Supplementary Fig. 1b), which is proven to be Sb precipitate[27]. The Sb precipitate will lead to additional crystal defects in this structure, such as Sb vacancies, while the $\alpha$-MgAgSb phase is maintained.

**INS measurements**. INS measurements were performed at the MARI time-of-flight chopper spectrometers at the ISIS Neutron and Muon Source in the UK[44]. Powder samples of 8.81645 and 9.97648 g for MgAg$_{0.97}$Sb$_{0.99}$ and MgAg$_{0.965}$-Ni$_{0.005}$Sb$_{0.99}$, respectively, were encased in a thin-walled aluminum cylinder of 40 mm diameter. The measurements were performed with incident neutron energies of $E_i = 50$ meV at 5 K using a closed cycle refrigerator (CCR). The background contributed by the CCR was subtracted. The data were subsequently combined to generate the GPDOS using the standard software MantidPlot[45]. Here, the generalized $Q$-dependent phonon density of states, $G(\mathbf{Q}, E)$, is related to the dynamic structure factor, $S(\mathbf{Q}, E)$, by the following equation[46,47],

$$G(\mathbf{Q}, E) = e^{Q^2 u^2} \left[ 1 - e^{-\frac{E}{k_B T}} \right] \frac{E}{Q^2} S(\mathbf{Q}, E)$$

where $\left[ 1 - e^{-\frac{E}{k_B T}} \right]$ describes the Bose–Einstein statistics, $e^{Q^2 u^2}$ describes the Debye–Waller factor, $u$ is the atomic thermal displacement, $k_B$ is the Boltzmann constant, and $T$ is the temperature. The values of $u$ at different temperatures are extracted from the powder diffraction data refinement. The neutron-weighted GPDOS values shown in Fig. 4a are the integration results of $G(\mathbf{Q}, E)$ with an integration $Q$ region of 2.7–6.9 Å$^{-1}$. The elastic peak is subtracted from the data <2.7 meV (energy resolution) and replaced with a monotonic function of energy that is characteristic of the inelastic scattering in the long-wavelength limit.

Multi-$E_i$ time-of-flight INS measurements were performed at the cold neutron disc chopper spectrometer BL14 AMATERAS[48] at the J-PARC in Japan. The chopper configurations were set to select $E_i$ of 42.00, 15.16, 7.74, and 4.68 meV at the low-resolution mode with high flux[49]. Powder samples of 6.2518 and 6.2399 g for MgAg$_{0.97}$Sb$_{0.99}$ and MgAg$_{0.965}$Ni$_{0.005}$Sb$_{0.99}$, respectively, were separately encased in a thin-walled aluminum cylinder. A top-loading closed cycle refrigerator (TL-CCR) was used for the temperature-dependent measurements. MgAg$_{0.97}$Sb$_{0.99}$ data were collected at 100, 300, 400, and 500 K, and MgAg$_{0.965}$Ni$_{0.005}$Sb$_{0.99}$ data are collected at 100, 200, 300, 400, and 500 K. The data reduction was completed using UTSUSEMI version 0.3.6. The background contributed by the TL-CCR was subtracted. The resulting $S(\mathbf{Q}, E)$, as a function of neutron energy transfer $E$ and momentum transfer $\mathbf{Q} = \mathbf{k_f} - \mathbf{k_i} = \mathbf{q} + \mathbf{\tau}$, where $\mathbf{k_i}$ and $\mathbf{k_f}$ are the incident and scattered neutron wave vector, respectively; $\mathbf{q}$ is the phonon wave vector; and $\mathbf{\tau}$ is the reciprocal lattice vector, was visualized in Mslice of the Data Analysis and Visualization Environment (DAVE)[50]. One-dimensional "$E$-cuts" were taken along the $Q$-axis to obtain the phonon spectra at specific $E$-points by DAVE. The Bose-factor-calibrated phonon dynamic structure factor, $B(\mathbf{Q}, E) = \left[ 1 - e^{-\frac{E}{k_B T}} \right] S(\mathbf{Q}, E)$, and the BPDOS were calculated by the Matlab program. The neutron-weighted BPDOS values shown in Fig. 4b are the integration results of $B(\mathbf{Q}, E)$ with a integration $Q$ region of 1.7–3.3 Å$^{-1}$. The peak positions of acoustic phonons and low-energy optical phonons were fitted by the OriginPro program using the Gaussian function.

**Computational methods**. The Kohn–Sham density-functional theory calculations[51,52] were performed using the projector-augmented-wave potential[53] within the generalized gradient approximation of the Perdew–Burke–Ernzerhof type[54] in the Vienna ab initio simulation package (VASP)[55]. During structural optimizations, all atomic positions and lattice parameters were fully relaxed until the maximum force allowed on each atom was <0.01 eVÅ$^{-1}$. The 0.03 Å$^{-1}$ spacing Monkhorst-Pack mesh with a cutoff of 500 eV was used in the calculations.

The vibrational properties of MgAgSb were studied within the harmonic approximation via density-functional perturbation theory (DFPT)[56] as implemented

in the Phonopy code[37] bundled with VASP. The harmonic properties have been performed on $2 \times 2 \times 2$ and $3 \times 3 \times 3$ supercells of $\alpha$- and $\gamma$-MgAgSb unit cells for DFPT calculations. In order to compare with experimental data of the measured multi-$E_i$ time-of-flight INS, the GPDOS of $\alpha$-MgAgSb was calculated by summing the partial phonon density of states (PhDOS) values weighted by the atomic scattering cross sections and masses:

$$GPDOS = \sum_i \frac{\sigma_i}{\mu_i} PhDOS_i$$

where $\sigma_i$ and $PhDOS_i$ represent the atomic scattering cross section and the PhDOS projected into the individual atoms, respectively. Here, the temperature-dependent phonon properties were calculated using the lattice parameters measured by neutron powder diffraction. The two-dimensional, $B(\mathbf{Q}, E)$ patterns shown in Fig. 3b, c were calculated from the ab initio phonon modes and polarization vectors with the OCLIMAX program[57].

The $\kappa_{lat}$ of MgAgSb here is directly determined from full solution of the phonon Boltzmann transport equation using the ShengBTE code[36]:

$$k_{lat} = \sum_\lambda^{3N} \int_q v_{i,q}^2 c_{i,q} \tau_{i,q} d\mathbf{q},$$

where $v_{i,q}$, $c_{i,q}$, and $\tau_{i,q}$ are the phonon group velocity, the mode specific heat capacity, and the relaxation time, respectively, for the $i$-th phonon mode at the wave vector $\mathbf{q}$ point. This process requires the energies and forces acting on a supercell for a set of random configurations generated by the trial density matrix. During the calculations, the same settings in ref. [38] were used for the anharmonic interatomic force constant calculations with large supercells of $2 \times 2 \times 2$ and $3 \times 3 \times 3$ for $\alpha$- and $\gamma$-MgAgSb unit cells, respectively.

## Data availability
The data that support the findings of this study are available from the corresponding authors upon reasonable request.

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

## Acknowledgements

X.Y.L. and F.W.W. acknowledge financial support by the National Natural Science Foundation of China (NSFC; no. 11675255) and the National Key R&D Program by the Ministry of Science and Technology (MOST) of China (no. 2016YFA0401503). X.-L.W. acknowledges financial support by the Croucher Foundation (CityU Project no. 9500034), the National NSFC (no. 51571170), and the National Key R&D Program by the MOST of China (no. 2016YFA0401501). H.Z.Z. acknowledges financial support by the MOST of China (no. 2018YFA0702100). This research used resources of the Advanced Photon Source, a U.S. Department of Energy (DOE) Office of Science User Facility operated for the DOE Office of Science by Argonne National Laboratory under Contract No. DE-AC02-06CH11357. The neutron scattering experiments were carried out at the ISIS/STFC facility, which is sponsored by the Newton-China fund (proposal RB1610190). The neutron scattering experiments at the CSNS were performed under user program (proposal no. P1819070300006). The neutron scattering experiments at the Materials and Life Science Experimental Facility (MLF), J-PARC were performed under user programs (AMATERAS proposal no. 2018A0061 and NOVA proposal no. 2018A0286).

## Author contributions

X.Y.L., E.Y.Z., M.K., K.N., and F.W.W. designed and performed the INS experiment using AMATERAS at the J-PARC. X.Y.L., T.G., M.D.L., and F.W.W. designed and performed the INS experiment using MARI at the ISIS. X.Y.L., Y.R., and X.L.W. designed and performed synchrotron X-ray diffraction measurements using 11-ID-C at the APS. X.Y.L., M.A., and F.W.W. designed and performed the high-resolution neutron diffraction experiment using Echidna at the ANSTO. X.Y.L., E.Y.Z., Z.G.Z., K.I., T.O., and F.W.W. designed and performed the neutron diffraction experiment using NOVA at the J-PARC. X.Y.L., E.Y.Z., J.C., L.H.H., and F.W.W. designed and performed the neutron diffraction experiment using GPPD at the CSNS. The samples were prepared by H.Z.Z. and Z.F.R. X.Y.L. analyzed the experiment data, along with all co-authors. P.F.L. and B.T.W. performed the ab initio calculations. X.Y.L., P.F.L., B.T.W., H.Z.Z., Z.F.R., and F.W.W. drafted the manuscript. All authors analyzed and reviewed the results, and provided input to this paper. F.W.W. conceived and supervised the project.

## Competing interests

The authors declare no competing interests.
