## [Peer Review File · Nature Communications]

Reviewers' comments:

Reviewer #1 (Remarks to the Author):

Recommendation: Publish after major revision.

In this manuscript entitled as "Ultralow Thermal Conductivity from Transverse Acoustic Phonon Suppression in Distorted Crystalline α -MgAgSb" by X. Li et. al. demonstrated the origins of low κ_{lat} microscopically using Neutron Diffraction, Inelastic Neutron Scattering (INS), Pair Distribution Function (PDF) and ab initio calculations. The authors debate that the main cause of such low κ_{lat} arises due to the suppression of Transverse acoustic phonons which is caused using distorted rock-salt lattice Mg-Sb lattice. The authors have used two different stoichiometric materials namely, MgAg_{0.97}Sb_{0.99} and MgAg_{0.965}Ni_{0.005}Sb_{0.99}. The authors have done a commendable job in understanding the cause for such low κ_{lat} , but at this present moment they need to answer few comments before acceptance in Nature Comm. The novelty of the work is high. However, if the authors could answer few of the queries listed below, it can improve the quality of the manuscript to be considered for publication.

1. The author mentions that "The structure features a 24-atom trigonal containing loosely bound atoms rattling in cages". However, rattling of an atom is possible only in cases where the cages are generally big for the guest fillers like in the cases of Skutterudites or Clathrates. Here in this distorted rock-salt structure. The tetrahedral hole does not fulfil such criteria of an atom to rattle. Also, the Ag being a considerably large atom will not be able to rattle freely in such tetrahedral hole. An atom being a rattler is often guided by high Atomic Displacement Parameters (ADPs). The Reitveld refinement shown here do not provide any information regarding the ADPs of atoms, their site occupancies which are crucial to assign an atom to the likes of rattler. Also, a characteristic for the presence of rattling atoms is a flat acoustic band which is not seen from the phonon spectrum. The authors might consider looking into this.
2. We can see a thermal dilation for both the samples with temperature. But the Ni-doped sample show a greater c/a ratio as compared to the undoped sample. The reason behind it is not explained. Also, the inflection point ~ 200 K is seen clearly for the undoped sample whereas for Ni-doped sample the inflection point seems to be at ~ 100 K. Also reference 23 shows an inflection point at around 300 K as is seen from Resistivity and Seebeck plot as well as from low temperature C_p data. I would like the authors to present diffraction data for 150 K and 250 K for removing this ambiguity regarding the inflection point.
3. The PDF data shown in figure 2a does not mention the temperature at which it is taken. Also, refinement of positions of atoms often leads to high correlation among themselves. The authors should mention clearly the fitting procedure done in PDFgui and how they have avoided the correlation among the atoms.
4. The author mentions that "applying them to the observed double-mushroom scattering pattern (Fig. 3a), it was determined that the lower branches at $E \sim 4.5$ meV are mainly the longitudinal acoustic phonon modes, whereas the upper branches at $E \sim 7.0$ meV are mainly the low-energy optical phonon modes." I feel that the author should mention the figure number (most probably referring to figure 4b) for this paragraph.
5. Why the authors chose Ni doped sample in particular along with the undoped sample. How did the authors confirm that Ni goes in Ag site. The Reitveld refinement again gives no indication of their position in this regard. The data shown from the Reitveld refinement can simply be obtained via

profile matching in FullProf. A complete table of atomic refinements and their position must be presented to augment the refined data.

6. Why Sb precipitates out at higher temperatures? If Ag is rattling, I would think that Ag might come out from the matrix due to their weak bonding. Furthermore, PhDOS reveals that Ag contributes to the low energy optical phonons which primarily scatters the transverse acoustic modes. In that case how static Mg off-centering from its mean position is responsible for ultralow κ .

7. In INS, chopper E_i of 42 meV was also used which would provide a total phonon DOS upto around 200 cm^{-1} or more. I would like the authors to provide data for full phonon DOS and compare with the ab initio results.

8. The 5 K and 100 K (and upwards) GPDOS data do not match. What is the reason behind this?

9. Author may mention few relevant reports on the analysis of ultralow thermal conductivity (ACS Energy Lett. 2018, 3, 1315; J. Am. Chem. Soc. 2018, 140, 5866 and J. Am. Chem. Soc. 2017, 139, 43503).

Reviewer #2 (Remarks to the Author):

The manuscript by X. Li et al investigates the lattice dynamics of MgAgSb for the first time experimentally using inelastic neutron diffraction, supported by phonon calculations. This work is timely, and the results will be interesting to a broad audience. However, I have concerns about the interpretation of the data and the reliance on powder measurements as opposed to use of single crystals. Further, the manuscript would benefit immensely from improved presentation and clarity.

Science aspects: I am not yet convinced that there is complete suppression of the transverse mode. In both of the simulated patterns (distorted and rock salt) the transverse modes are very faint. The authors should comment on the use of powder samples to obtain the full dispersion, and the limitations of this approach in comparison to using single crystals. For comparison, can they point to related examples in which powder data has yielded dispersions with clear transverse modes?

Formatting and clarity:

-Missing section headings. There are no headings for "results and discussion" or "conclusions".

-As written, the article is very difficult and frustrating to wade through. The figures are not clearly presented, especially Fig 3a-d. If the authors used a narrower E-Q range, they may be able to clearly label the different elements and help the reader understand.

- Figure 2: I would suggest using color coding so that the crystal structure bonds match the corresponding PDF data. Currently, the reader is forced to match the bonds with the atoms one by one.

In Figure 3 caption, the authors state "The magenta and green lines are calculated dispersions based on transverse and longitudinal sound velocities, respectively, which were measured by the resonant ultrasound spectrometer method". These green and magenta lines are difficult to see, and they are not labeled.

Reviewer #3 (Remarks to the Author):

This paper is premature to be published. I was expecting to see figure showing comparison of experimental and theoretical thermal conductivity data based on authors finding. Without this figure, authors cannot claim 'Ultralow thermal conductivity from transverse acoustic phonon suppression' If authors want to update the manuscript by including 'the figure', I would like to see which effect is dominant; suppression of phonon group velocity owing to suppression in transverse phonon or changes in anharmonic scattering due to phonon dispersion changes. Also, it would be very useful to community phonon dispersion measurement based on neutron inelastic scattering.

Reviewers' comments:

Reviewer #1 (Remarks to the Author):

Recommendation: Publish after major revision.

In this manuscript entitled as “Ultralow Thermal Conductivity from Transverse Acoustic Phonon Suppression in Distorted Crystalline α -MgAgSb” by X. Li et. al. demonstrated the origins of low k_{lat} microscopically using Neutron Diffraction, Inelastic Neutron Scattering (INS), Pair Distribution Function (PDF) and ab initio calculations. The authors debate that the main cause of such low k_{lat} arises due to the suppression of Transverse acoustic phonons which is caused using distorted rock-salt lattice Mg-Sb lattice. The authors have used two different stoichiometric materials namely, $MgAg_{0.97}Sb_{0.99}$ and $MgAg_{0.965}Ni_{0.005}Sb_{0.99}$. The authors have done a commendable job in understanding the cause for such low k_{lat} , but at this present moment they need to answer few comments before acceptance in Nature Comm. The novelty of the work is high. However, if the authors could answer few of the queries listed below, it can improve the quality of the manuscript to be considered for publication.

Response: We thank Referee #1 for his/her careful review of our work, and for providing insightful comments that helped further improve our manuscript. We are delighted that the Referee #1 considers that our work “is a commendable job in understanding the cause for such low k_{lat} ” and “the novelty of the work is high”. A detailed response to all comments is listed below.

1. The author mentions that “The structure features a 24-atom trigonal containing loosely bound atoms rattling in cages”. However, rattling of an atom is possible only in cases where the cages are generally big for the guest fillers like in the cases of Skutterudites or Clathrates. Here in this distorted rock-salt structure. The tetrahedral hole does not fulfil such criteria of an atom to rattle. Also, the Ag being a considerably large atom will not be able to rattle freely in such tetrahedral hole. An atom being a rattler is often guided by high Atomic Displacement Parameters (ADPs). The Reitveld refinement shown here do not provide any information regarding the ADPs of atoms, their site occupancies which are crucial to assign an atom to the likes of rattler. Also, a characteristic for the presence of rattling atoms is a flat acoustic band which is not seen from the phonon spectrum. The authors might consider looking into this.

Response: We thank Referee #1 for pointing out this issue. It is difficult to refine all the atomic displacement parameters and occupancy parameters simultaneously due to their strong correlations and data quality dependence. We tried to refine the atomic displacement parameters under the constraints of all Ag atoms have the same thermal displacements and all atom occupancies are fixed. The refined atomic displacement parameters are shown in

Table.r1. The thermal displacements of Ag atoms are smaller than other atoms which reflects their non-rattling nature. We agree that there is no flat acoustic band in the phonon spectrum (Fig. S2a). Thus, we totally agree with Referee #1 that the term “rattling” is baseless which may confuse readers. We have deleted the statement of “*containing loosely bound atoms “rattling” in cages*” in the revised manuscript.

Two tables (Table S1 and Table S2) including the refinement parameters are added in the supplementary information in the revised manuscript.

Table r1. Atomic displacement parameters obtained from Rietveld refinement.

T/K	U_{iso_Mg} (Å ²)*	U_{iso_Ag} (Å ²)	U_{iso_Sb} (Å ²)	R_{wp} (%)	χ^2
MgAg_{0.97}Sb_{0.99}:					
20	0.41(6)	0.04(4)	0.29(6)	1.10	1.93
50	0.44(6)	0.09(4)	0.25(6)	1.39	2.00
100	0.47(6)	0.10(4)	0.27(6)	0.71	1.99
200	0.58(7)	0.13(4)	0.45(6)	1.52	1.82
300	0.63(7)	0.62(5)	0.63(6)	2.83	1.89
350	0.85(8)	0.67(6)	0.77(6)	2.65	1.93
400	1.25(9)	0.91(6)	1.07(6)	1.04	1.76
450	1.56(9)	1.03(6)	1.26(6)	1.82	1.90
500	1.60(1)	1.41(6)	1.33(7)	2.89	2.04
MgAg_{0.965}Ni_{0.005}Sb_{0.99}:					
20	0.40(6)	0.04(4)	0.29(5)	0.77	2.65
50	0.41(6)	0.09(4)	0.26(5)	1.54	2.76
100	0.45(6)	0.07(4)	0.30(6)	1.29	2.85
200	0.52(7)	0.21(4)	0.41(6)	1.88	2.74
300	0.63(7)	0.47(5)	0.80(6)	2.50	2.56
350	0.79(7)	0.79(5)	1.08(7)	1.49	2.83
400	1.07(7)	0.92(5)	1.29(7)	3.27	3.00
450	1.21(8)	1.15(6)	1.51(7)	3.12	3.09
500	1.62(9)	1.21(6)	1.58(7)	3.77	3.33

* Here, $U_{iso} = 8\pi^2\mu^2$, where μ^2 is the root mean square deviation of atoms from its equilibrium position in Å².

2. We can see a thermal dilation for both the samples with temperature. But the Ni-doped sample show a greater c/a ratio as compared to the undoped sample. The reason behind it is not explained. Also, the inflection point ~200 K is seen clearly for the undoped sample whereas for Ni-doped sample the inflection point seems to be at ~100 K. Also reference 23 shows an inflection point at around 300 K as is seen from Resistivity and Seebeck plot as well as from low temperature C_p data. I would like the authors to present diffraction data for 150 K and 250 K for removing this ambiguity regarding the inflection point.

Response: We thank Referee #1 for pointing out these comments. The crystal ionic radii of Ag⁺ and Ni²⁺ are 1.29 Å and 0.83 Å [R. D. Shannon. *Acta Crystallogr A*. **32**(5), 751 (1976)]. Thus, the Ni doping can significantly shrink the a/b-axis while cannot influence the c-axis that much as shown in Fig. r1. This is why the Ni-doped sample shows a greater c/a ratio

as compared to the undoped one. Here, we propose the anisotropic shrink is related to the smaller Ni^{2+} radius and shorter bond length between Ni and Sb. We have added a statement to clarify, “*we propose that this anisotropic shrink is related to the smaller Ni^{2+} radius and shorter bond length between Ni and Sb*” in the caption of Fig.S1 in the revised manuscript.

We thank Referee #1 for suggesting to us to carry out further diffraction measurements for presenting 150 and 250 K data. We have applied the rapid access neutron beam time at China Spallation Neutron Source (CSNS) and obtained the data for both $\text{MgAg}_{0.97}\text{Sb}_{0.99}$ and $\text{MgAg}_{0.965}\text{Ni}_{0.005}\text{Sb}_{0.99}$ samples. Results are presented in Fig. S1 in the new version of manuscript. The results show more clearly that the inflection point is ~ 200 K for the $\text{MgAg}_{0.97}\text{Sb}_{0.99}$ and a little bit lower ~ 100 K for $\text{MgAg}_{0.965}\text{Ni}_{0.005}\text{Sb}_{0.99}$ (see Fig. r1).

We previously proved that the inflection points of the resistivity and Seebeck coefficient in reference 23 (revised manuscript Ref. 27) are as a result of the bipolar effect [Li, D. *et al. Adv. Funct. Mater.* **25**, 6478–6488 (2015)]. The inflection points reported in the total thermal conductivity and ZT around 100 to 200 K may consistent with the inflection points of the lattice parameters shown in the present work. We propose that additional phonon-scattering mechanism related to the structure evolution may appear at high temperatures.

Fig. r1. The comparison of refined lattice parameters as a function of temperature in $\text{MgAg}_{0.97}\text{Sb}_{0.99}$ and $\text{MgAg}_{0.965}\text{Ni}_{0.005}\text{Sb}_{0.99}$. The data show that the Ni doping can significantly shrink the a/b -axis while cannot influence the c -axis that much.

3. The PDF data shown in figure 2a does not mention the temperature at which it is taken. Also, refinement of positions of atoms often leads to high correlation among themselves. The authors should mention clearly the fitting procedure done in PDFgui and how they have avoided the correlation among the atoms.

Response: We are sorry for not making the caption clear. In this work, the PDF data shown in Fig.2a is measured at 300 K. We have added a statement to clarify, “*The data is measured at 300 K*” in the caption of Fig.2a in the revised manuscript.

We agree with the Referee #1 that the PDF data refinement of positions of atoms often leads to high correlation among themselves. In our case, the PDF data revealed the static local structure distortion, and the positions of all atoms in unit cell are written and refined. The symmetry constrains are generated by the symmetry of the space group. The values of instrument resolution dampening factor Q_{damp} and resolution peak broadening factor Q_{broad} are determined from standard Si powder data and are fixed in the refinement. We have added a statement of “*In the refinement, the positions of all atoms in unit cell are written and refined. The symmetry constrains are generated by the symmetry of the space group. The values of instrument resolution dampening factor Q_{damp} and resolution peak broadening factor Q_{broad} are determined from standard Si powder data and are fixed in the refinement*” in the methods in the revised manuscript.

4. The author mentions that “applying them to the observed double-mushroom scattering pattern (Fig. 3a), it was determined that the lower branches at $E \sim 4.5$ meV are mainly the longitudinal acoustic phonon modes, whereas the upper branches at $E \sim 7.0$ meV are mainly the low-energy optical phonon modes.” I feel that the author should mention the figure number (most probably referring to figure 4b) for this paragraph.

Response: We agree that this is not stated clearly. We have added a statement to mention the figure numbers “...whereas the upper branches at $E \sim 7.0$ meV are mainly the low-energy optical phonon modes (Fig. 3a and Fig. 4b)” in the revised manuscript.

5. Why the authors chose Ni doped sample in particular along with the undoped sample. How did the authors confirm that Ni goes in Ag site. The Reitveld refinement again gives no indication of their position in this regard. The data shown from the Reitveld refinement can simply be obtained via profile matching in FullProf. A complete table of atomic refinements and their position must be presented to augment the refined data.

Response: We thank Referee #1 for these insightful comments. The Ni *p*-type substitution for Ag was intended to decrease the electrical resistivity, but our previous results showed that the Ni-doping decreases the carrier mobility leading to higher electrical resistivity, though the thermal conductivity was decreased, consequently leading to enhanced *ZT* for the $\text{MgAg}_{0.965}\text{Ni}_{0.005}\text{Sb}_{0.99}$ alloy [Zhao, H. et al. *Nano Energy* 7, 97–103 (2014)]. Here, we designed these experiments to study this anomalous property. In view of this insightful comment, we have added a statement, “(Ni *p*-type substitution for Ag sample has an anomalous higher electrical resistivity)” to highlight the aim of study Ni-doped sample in the introduction part in the revised manuscript. The results are discussed in the discussion part.

For the response of the confirmation of Ni going in Ag site. Here, the $\sim 0.5\%$ Ni doping cannot be determined by the Rietveld refinement limited by the resolution. To address this comment, we further performed Hall measurements and determined the carrier

concentrations (n_H) (Table r2). The n_H is increased more than 10% after Ni doping, which demonstrated that the Ni^{2+} doped in Ag^{1+} site instead of Mg^{2+} . While the resistivity increases from the decreasing of carrier mobility. These data are added in the supplementary information in the revised manuscript (Table S4).

Two tables (Table S1 and Table S2) including the refinement parameters are also added in the supplementary information in the revised manuscript.

Table r2. Resistivity, carrier mobility and carrier concentrations (n_H) determined from Hall measurements at room-temperature.

	Resistivity ($\Omega \cdot \text{cm}$)	Mobility ($\text{cm}^2/\text{V/s}$)	n_H ($10^{19}/\text{cm}^3$)
MgAg_{0.97}Sb_{0.99}:			
Heat annealing	0.002054	77.6	+3.916
	0.002054	74.4	+4.086
	0.002058	76.2	+3.979
Mean value	0.002056	76.1	+3.994
MgAg_{0.965}Ni_{0.005}Sb_{0.99}:			
Heat annealing	0.003082	49.1	+4.121
	0.003083	45.0	+4.498
	0.003080	43.6	+4.652
Mean value	0.003082	45.9	+4.424

6. Why Sb precipitates out at higher temperatures? If Ag is rattling, I would think that Ag might come out from the matrix due to their weak bonding. Furthermore, PhDOS reveals that Ag contributes to the low energy optical phonons which primarily scatters the transverse acoustic modes. In that case how static Mg off-centering from its mean position is responsible for ultralow k_{lat} .

Response: We thank Referee #1 for raising this interesting comment. We explained that the Ag atoms are not rattling in the reply of comment 1. Actually, this MgAgSb sample is very difficult synthesized to single-phase and we found a two-step process method combining ball milling with hot pressing for synthesizing the single-phase sample [Zhao, H. et al. *Nano Energy* 7, 97 (2014)]. Many previous works show that the Sb and Ag_3Sb are precipitated simultaneously due to charge-balancing, and the Ag_3Sb impurity peaks are always weaker than the Sb impurity peaks [Kirkham, M. J. et al. *Phys. Rev. B* 85, 144120 (2012); Zhao, H. et al. *Nano Energy* 7, 97 (2014); Ying, P. et al. *Chem. Mater.* 27, 909 (2015)]. In our case, the sample is single-phase at room-temperature and there are tiny Sb precipitates at temperatures above 450 K, and the Ag_3Sb precipitates diffraction peaks are too weak to be measured.

The PhDOS reveals that the low energy optical phonons (5~7 meV in Fig.S6a) are mainly contributed by Ag and Sb atoms. In α -MgAgSb phase, the Mg and Sb atoms are all off-centered (shown in Table S1). The Mg-Sb rocksalt-type sublattice distortion caused the

distortion of the bonds related to Ag atoms. Here, we demonstrated that the distortion structure has a significant scattering on transverse acoustic modes. That's how the distortion structure responsible for the ultralow k_{lat} .

7. In INS, chopper E_i of 42 meV was also used which would provide a total phonon DOS upto around 200 cm^{-1} or more. I would like the authors to provide data for full phonon DOS and compare with the *ab initio* results.

Response: We agree in principle with the Referee #1 that the chopper E_i of 42 meV data can provide a full phonon DOS which can be compared with the *ab initio* results. Here, we didn't show these data due to the poor resolution. The comparison data between INS measured DOS and *ab initio* calculated DOS has been shown in Fig. 4a. The total phonon DOS data in this figure are measured at MARI@ISIS.

8. The 5 K and 100 K (and upwards) GPDOS data do not match. What is the reason behind this?

Response: In the current manuscript, the 5 K GPDOS data (Fig. 4a) are measured at MARI@ISIS with $E_i = 50.00$ meV (low energy resolution) aimed for the full DOS measurements. However, the 100 to 500 K data (Fig. 4b) are measured at AMATERAS@J-PARC with $E_i = 15.16$ meV (high energy resolution) aimed for the accurate measurements of the low energy acoustic modes. Thus, the high resolution 100 to 500 K data show the separation between the longitudinal acoustic phonon modes (~ 4.5 meV) and the low-energy optical phonon modes (~ 7.0 meV), which is not measured at MARI@ISIS (Fig. 3a and Fig. 4a&b).

9. Author may mention few relevant reports on the analysis of ultralow thermal conductivity (ACS Energy Lett. 2018, 3, 1315; J. Am. Chem. Soc. 2018, 140, 5866 and J. Am. Chem. Soc. 2017, 139, 43503).

Response: We thank Referee #1 for providing useful information about the analysis of ultralow thermal conductivity. We have added the references 'ACS Energy Lett. 2018, 3, 1315 [ref. 11]; J. Am. Chem. Soc. 2018, 140, 5866 [ref. 24]; J. Am. Chem. Soc. 2017, 139, 4350 [ref. 19]; and Energy Environ. Sci. 2019, 12, 589 [ref. 20]' in the revised manuscript.

Reviewer #2 (Remarks to the Author):

The manuscript by X. Li et al investigates the lattice dynamics of MgAgSb for the first time experimentally using inelastic neutron diffraction, supported by phonon calculations. This work is timely, and the results will be interesting to a broad audience. However, I have concerns about the interpretation of the data and the reliance on powder measurements as opposed to use of single crystals. Further, the manuscript would benefit immensely from improved presentation and clarity.

Response: We thank Referee #2 for his/her critical and helpful comments, and for stating that our work is “*timely and the results will be interesting to a broad audience*”. We have taken the Referee’s suggestions very seriously and have performed a major revision. Below, we addressed all the questions raised by the reviewer:

Science aspects: I am not yet convinced that there is complete suppression of the transverse mode. In both of the simulated patterns (distorted and rock salt) the transverse modes are very faint. The authors should comment on the use of powder samples to obtain the full dispersion, and the limitations of this approach in comparison to using single crystals. For comparison, can they point to related examples in which powder data has yielded dispersions with clear transverse modes?

Response: The Referee #2 raises an important point. We fully agree that the limitations of phonon dispersion measurements on powder sample in comparison to using single crystals. In this work, we studied powder sample as we cannot synthesize the single crystal of this material till now. The effectiveness of the phonon measurements with clear transverse modes on powder samples (see Fig. r2) can be appreciated in recent studies published by *Li et al* [*Nat. Mater.* **17**, 226 (2018)] which has been cited in our manuscript [Ref. 1]. There, the mechanism of liquid-like thermal conduction in AgCrSe₂ lies in the transverse acoustic phonons suppression by the dynamic disorder with crystal structure transition. Another example of phonon studying on powder sample is published by *Niedziela et al* [*Nat. Phys.* **15**, 73–78 (2019)] which has been cited in our manuscript [Ref. 2]. They measured acoustic phonons in powder sample (see Fig. r3) and the mechanism of low thermal conduction in AgCuSe₂ is the anharmonic phonon dynamics.

In response to this comment, the Fig. 3 is revised which made the longitudinal and transverse phonon modes more clearly in the revised manuscript.

Fig. r2. (a) The inelastic neutron scattering (INS) data show transverse acoustic (TA) modes in powder AgCrSe_2 sample at low temperature and (b) the INS data show TA suppressed at high temperature. (c, d) Contour plot of $S(Q, E)$ as a function of temperature and the half-width at half-maximum of the TA phonons and the diffuse scattering determined in spectral fitting. (For more details, *see Ref. 1, Li et al, Nat. Mater. 17, 226 (2018)*).

Fig. r3. (a) Room-temperature crystal structure of CuCrSe_2 . (b-d) The temperature-dependent INS data show the dispersive acoustic phonons emanating from the (110) Bragg peak near 3.4\AA^{-1} in powder CuCrSe_2 sample. (e) High-temperature structure shown with Cu occupancy on α and β sites (indicated with half white-half red spheres). (For more details, *see Ref. 2, Niedziela et al, Nat. Phys. 15, 73–78 (2019)*).

Formatting and clarity:

-Missing section headings. There are no headings for “results and discussion” or “conclusions”.

Response: We are sorry for missed these headings. We have added “*Results*” and “*Discussion*” in the revised manuscript. The manuscript is reformatted to complies with Nature Communications editorial policies.

-As written, the article is very difficult and frustrating to wade through. The figures are not clearly presented, especially Fig 3a-d. If the authors used a narrower E-Q range, they may be able to clearly label the different elements and help the reader understand.

Response: We thank Referee #2 for pointing out this issue. We have revised Fig. 3 based on this informative comment in the revised manuscript. We magnified the areas of $B(Q, E)$ at the Q from 2.2 to 3.3 \AA^{-1} and E from 0 to 10 meV in Figs. 3a-3c, making their differences clearly visible. We have added a statement of “*Magnifications of areas of $B(Q, E)$ at the Q from 2.2 to 3.3 \AA^{-1} and E from 0 to 10 meV region in Figs. 3a-3c are shown in Fig. S4, making their differences clearly visible*” in the results and showed the magnified area of $B(Q, E)$ in Fig. S4 in the supplementary information in the revised manuscript.

- Figure 2: I would suggest using color coding so that the crystal structure bonds match the corresponding PDF data. Currently, the reader is forced to match the bonds with the atoms one by one.

Response: We agree with Referee #2 that the color-coding method is very useful for reader friendly. In response to this comment, we have revised Fig. 2 using color-coding in the revised manuscript. In our case, aims for highlight the Mg-Sb sublattice distortion, only the Mg-Sb bonds are shown in Fig. 2b.

- In Figure 3 caption, the authors state “The magenta and green lines are calculated dispersions based on transverse and longitudinal sound velocities, respectively, which were measured by the resonant ultrasound spectrometer method”. These green and magenta lines are difficult to see, and they are not labeled.

Response: We have revised Fig. 3 based on this informative comment in the revised manuscript.

Reviewer #3 (Remarks to the Author):

This paper is premature to be published. I was expecting to see figure showing comparison of experimental and theoretical thermal conductivity data based on authors finding. Without this figure, authors cannot claim 'Ultralow thermal conductivity from transverse acoustic phonon suppression' If authors want to update the manuscript by including 'the figure', I would like to see which effect is dominant; suppression of phonon group velocity owing to suppression in transverse phonon or changes in anharmonic scattering due to phonon dispersion changes. Also, it would be very useful to community phonon dispersion measurement based on neutron inelastic scattering.

Response: We thank Referee #3 for raising this insightful point. We fully agree that the comparison of experimental and theoretical thermal conductivity data will increase the persuasive power of our viewpoint. The experimental lattice thermal conductivity (k_{lat}) was calculated by subtracting the electronic thermal conductivity (k_{ele}) from total thermal conductivity (k_{tot}). k_{ele} was obtained using the Wiedemann-Franz law: $k_{\text{ele}} = L\sigma T$, where L is the Sommerfeld Lorentz number in the range from 1.6×10^{-8} to 2.6×10^{-8} $\text{W}\Omega\text{K}^{-2}$ [Baroni, S. et al, *Rev. Mod. Phys.* **73**, 515–562 (2001); Zhao, H. et al. *Nano Energy* **7**, 97–103 (2014)]. The theoretical k_{lat} was calculated by solving the linearized Boltzmann transport equation with the ShengBTE package [Li, W. et al, *Comput. Phys. Commun.* **185**, 1747–1758 (2014)]:

$$k_{\text{lat}} = \sum_{\lambda}^{3N} \int_{\mathbf{q}} v_{i,\mathbf{q}}^2 c_{i,\mathbf{q}} \tau_{i,\mathbf{q}} d\mathbf{q},$$

where $v_{i,\mathbf{q}}$, $c_{i,\mathbf{q}}$, and $\tau_{i,\mathbf{q}}$ are the phonon group velocity, the mode specific heat capacity, and the relaxation time, respectively, for the i -th phonon mode at the wave-vector \mathbf{q} point. In Fig. r4, it is observed that the predicted values are in good agreement with the measured values. The calculated room-temperature k_{lat} of α -MgAgSb is $0.54 \text{ Wm}^{-1}\text{K}^{-1}$, which is comparable with the experimental value of MgAg_{0.97}Sb_{0.99} ($\sim 0.6 \text{ Wm}^{-1}\text{K}^{-1}$) [Zhao, H. et al. *Nano Energy* **7**, 97–103 (2014)]. As clearly indicated in Figs. r4b-r4c, the phonon group velocities of α -MgAgSb are lower than those of the high-symmetry structure γ -MgAgSb, while its phonon lifetimes are larger than those of γ -MgAgSb, especially for the acoustical phonon frequency below 10 meV in γ -MgAgSb. These modes could make the three-phonon process easier to occur (Fig. r5) and accordingly enhance the phonon anharmonicity of γ -MgAgSb. The total Grüneisen parameter (γ_{total}) obtained as a weighted sum of the mode contributions at 300 K are 1.51 and 3.03 for α -MgAgSb and γ -MgAgSb, respectively (Fig. r4d). Generally, large γ_{total} corresponds to large phonon anharmonicity and low k_{lat} , such as the γ_{total} are 1.45 for PbTe, 2.83 for SnS, 3.5 for AgSbSe, and 3.9 for CsAg₅Te, corresponding to measured room-temperature k_{lat} of 2.4, 0.62, 0.48, and 0.2 $\text{Wm}^{-1}\text{K}^{-1}$, respectively [Xiao, Y. et al. *Phys. Rev. B* **94**, 125203 (2016); Nielsen, M. D. et al. *Energy Environ. Sci.* **6**, 570–578 (2013); Lin, H. et al. *Angew. Chemie Int. Ed.* **55**, 11431–11436 (2016)]. Such an $\gamma_{\text{total}} \sim 1.51$ of α -MgAgSb should therefore give a much larger k_{lat} than that in anomalously

high anharmonicity γ -MgAgSb ($\gamma_{\text{total}} \sim 3.03$), although its v are slightly lower than those of α -MgAgSb. However, our calculations indicate that the k_{lat} of α -MgAgSb and γ -MgAgSb are nearly equal over 300 K with value differences below $0.06 \text{ Wm}^{-1}\text{K}^{-1}$ (Fig. r4a and Fig. S8). Meanwhile, the average Grüneisen parameter γ_{total} of γ -MgAgSb ($\gamma_{\text{total}} \sim 3.03$) is much higher than the value of α -MgAgSb ($\gamma_{\text{total}} \sim 1.51$). This comes as no surprise, since the two polymorphisms follow with different low thermal conductivity mechanisms: phonon scattering resulting from the structure distortion in the weak-anharmonicity (low γ_{total}) α -MgAgSb and giant phonon anharmonicity (high γ_{total}) in γ -MgAgSb.

In the result part, one paragraph with Fig. 5 is added in the revised manuscript. Two figures show the calculated three-phonon scattering phase space and lattice thermal conductivity (Fig. S7 and Fig. S8) are added in the supplementary information in the revised manuscript.

The added paragraph is “**Low thermal conductivity mechanism.** To further verify the low thermal conductivity mechanism, we computed the intrinsic anharmonic effects of α -MgAgSb and high-symmetry structure MgAgSb (e.g. the γ -MgAgSb shown in Fig. 3f) from first principles using ShengBTE³⁶ and Phonopy³⁷. Figure 5a shows temperature-dependent k_{lat} of α -MgAgSb and it demonstrates overall agreement with our experimental results of MgAg_{0.97}Sb_{0.99}^{15,38}. The calculated room-temperature k_{lat} of α -MgAgSb is $0.54 \text{ Wm}^{-1}\text{K}^{-1}$, which is comparable with the experimental value of MgAg_{0.97}Sb_{0.99} ($\sim 0.6 \text{ Wm}^{-1}\text{K}^{-1}$)¹⁵. As clearly indicated in Fig. 5b and Fig. 5c, the phonon group velocities (v) of α -MgAgSb are lower than those of the high-symmetry structure γ -MgAgSb, while its phonon lifetimes (τ) are larger than those of γ -MgAgSb, especially for the acoustical phonons below 10 meV in γ -MgAgSb. These modes could make the three-phonon process easier to occur (Fig. S7) and accordingly enhance the phonon anharmonicity of γ -MgAgSb. The total Grüneisen parameter (γ_{total}) obtained as a weighted sum of the mode contributions at 300 K are 1.51 and 3.03 for α -MgAgSb and γ -MgAgSb, respectively (Fig. 5d). Generally, large γ_{total} corresponds to large phonon anharmonicity and low k_{lat} , such as the γ_{total} are 1.45 for PbTe³⁹, 2.83 for SnSe³⁹, 3.5 for AgSbSe₂⁴⁰, and 3.9 for CsAg₅Te₃⁴¹, corresponding to measured room-temperature k_{lat} of 2.4, 0.62, 0.48, and 0.2 $\text{Wm}^{-1}\text{K}^{-1}$, respectively. Such an $\gamma_{\text{total}} \sim 1.51$ of α -MgAgSb should therefore give a much larger k_{lat} than that in anomalously high anharmonicity γ -MgAgSb ($\gamma_{\text{total}} \sim 3.03$), although its v are slightly lower than those of γ -MgAgSb. However, our calculations indicate that the k_{lat} of α -MgAgSb and γ -MgAgSb are nearly equal over 300 K with value differences below $0.06 \text{ Wm}^{-1}\text{K}^{-1}$ (Fig. 5a and Fig. S8). This, from a side, confirms our hypothesis that the fully scattered transverse acoustic phonons by the static local structure distortion greatly reduce k_{lat} of the weak-anharmonicity α -MgAgSb and make it comparable with giant anharmonic materials²³.”

With this response letter to address Referee #3’s concerns and the revised manuscript, we hope that Referee #3 would agree that the combined neutron scattering data and *ab initio* calculation results are sufficient to support the statement of ultralow thermal conductivity

from transverse acoustic phonon suppression in α -MgAgSb, which is a surprise finding and important for understanding and designing high-performance thermoelectric materials.

Fig. r4 | Phonon Transport Properties in α -MgAgSb and in the high-symmetry structure MgAgSb (e.g. the γ -MgAgSb shown in Fig. 3f). (a), Experimental and theoretical temperature-dependent thermal conductivity of the α -MgAgSb phase. The literature values from Ref. 38 (Cal.) and Ref. 15 (Exp.) are also plotted for comparison. According to $k_{\text{tot}} = k_{\text{lat}} + k_{\text{ele}} = k_{\text{lat}} + L\sigma T$, the lattice thermal conductivity (k_{lat}) can be obtained by subtracting k_{ele} from the k_{tot} . The shadow regions are the experimental thermal conductivity of $\text{MgAg}_{0.97}\text{Sb}_{0.99}$ with the Sommerfeld Lorenz number^{15,56} being from 1.6×10^{-8} to $2.6 \times 10^{-8} \text{ W}\Omega\text{K}^{-2}$. Here, we only deal with the pure α -MgAgSb crystal and consider the phonon-phonon coupling to simulate the phonon transport properties. Our calculated values are in excellent agreement with the previous reports and our measured values^{15,38}. (b), Calculated phonon group velocities v , (c), phonon relaxation time τ , and (d), Grüneisen parameters γ_{gru} for α - and γ -MgAgSb at 300 K. The inset in (d) shows total γ_{total} obtained as a weighted sum of the mode contributions as a function of temperature for α - and γ -MgAgSb. The γ_{gru} of high value accumulate in the vicinity of 5 meV corresponding to the transverse acoustic phonon modes for γ -MgAgSb.

Fig. r5 | Calculated three-phonon scattering phase space of α -MgAgSb and in the high-symmetry structure MgAgSb (e.g. the γ -MgAgSb shown in Fig. 3f). The average three-phonon scattering phase space of α -MgAgSb is much lower than that of γ -MgAgSb, especially for the transverse acoustic modes and longitudinal acoustic modes in the low frequency region (<12 meV). This greatly weakens the intrinsic phonon-phonon scattering, increases the phonon lifetime of these phonon modes (Fig. 5c), and gives rise to the decrease of phonon anharmonicity of γ -MgAgSb. However, the static local structure distortion fully scattered the transverse acoustic phonons and accordingly suppress the phonon transport in α -MgAgSb. As a result, the mechanisms of static local structure distortion suppression transverse phonons in α -MgAgSb and giant anharmonicity in γ -MgAgSb lead to the nearly equal k_{lat} of the two polymorphisms.

Reviewers' comments:

Reviewer #1 (Remarks to the Author):

Author have satisfactorily answered comments of both the reviewers. This paper can be accepted as is.

Reviewer #2 (Remarks to the Author):

The authors have addressed my primary concern regarding interpretation of the powder inelastic scattering data. The figures have been improved for clarity. In my opinion, the manuscript can be published in its current state.

Reviewer #3 (Remarks to the Author):

I found the updated manuscript is mostly consistent and almost ready to be published. One comment I have is

although authors have demonstrated suppression in TA phonon, arguments saying 'three phonon processes are easier to occur' needs to be validated. In Figure 5a, deviation between experimental data and theoretical analysis gets larger at higher temperature suggesting U-process is overestimated in simulation.

Authors should add comments on this then the paper should be fully consistent.

Reviewers' comments:

Reviewer #1 (Remarks to the Author):

Author have satisfactorily answered comments of both the reviewers. This paper can be accepted as is.

Response: We thank Reviewer #1 for his/her second careful review of our work, and for this high evaluation on our revised manuscript.

Reviewer #2 (Remarks to the Author):

The authors have addressed my primary concern regarding interpretation of the powder inelastic scattering data. The figures have been improved for clarity. In my opinion, the manuscript can be published in its current state.

Response: We thank Reviewer #2 for his/her second careful review of our work, and for this high evaluation on our revised manuscript.

Reviewer #3 (Remarks to the Author):

I found the updated manuscript is mostly consistent and almost ready to be published. One comment I have is

although authors have demonstrated suppression in TA phonon, arguments saying 'three phonon processes are easier to occur' needs to be validated. In Figure 5a, deviation between experimental data and theoretical analysis gets larger at higher temperature suggesting U-process is overestimated in simulation.

Authors should add comments on this then the paper should be fully consistent.

Response: We thank Reviewer #3 for his/her second careful review of our work, and for providing this insightful comment that helped further improve our manuscript. Below, we addressed it very seriously:

We apologize for not being the statement of '*three-phonon processes are easier to occur*' clear before in the last revised manuscript. To eliminate possible misunderstandings, here, we restructuring our writing logical:

1, The calculation of three-phonon scattering phase space shown in Fig. S7 demonstrated that the three-phonon process is easier to occur in γ -MgAgSb compared to α -MgAgSb (methods from Lindsay, L. & Broido, D. A. J. *Phys. Condens. Matter* **20**, 165209 (2008), and Lee, S. et al. *Nat. Commun.* **5**, 3525 (2014)). This explains why α -MgAgSb has a longer relaxation time (Fig. 5c) and weaker phonon anharmonicity (Fig. 5d) than γ -MgAgSb. The weaker phonon anharmonicity (Fig. 5d) in α -MgAgSb should, therefore, give a larger k_{lat} than in γ -MgAgSb. However, our calculations indicate that the k_{lat} of α - and γ -MgAgSb are nearly equal over 300 K (Fig. 5a and Fig. S8). This, from a side, confirms our hypothesis that the TA phonon suppression by the static local structure distortion greatly reduce k_{lat} in α -MgAgSb, and make it comparable with k_{lat} in γ -MgAgSb where giant anharmonic phonon (large Grüneisen parameter $\gamma_{\text{total}} \sim 3.03$) is the main low k_{lat} mechanism.

2, We totally agree that the deviation between experimental data and theoretical analysis gets larger at higher temperature in Fig. 5a suggesting U-process is overestimated in simulation. Yet, from first principles, it is extremely difficult to quantify their accurate contribution to thermal transport. Also, from experiments, k_{lat} was obtained by subtracting k_{ele} from k_{tot} which depends on the value of Lorenz number (Fig. 5a). To confirm our computational methods, the literature calculation values from Ref. 38 have been plotted in Fig. 5a for comparison which are highly consistent with our calculations. Meanwhile, although overestimation exists, it does not affect the main idea of this manuscript: the TA phonon suppression by distorted rocksalt structure induces ultralow thermal conductivity in α -MgAgSb. Here, the TA phonon suppression is demonstrated by the fact that the INS and simulation data of α -MgAgSb all show TA phonon suppression (Fig. 3a and Fig. 3b), while the simulation of non-distorted γ -MgAgSb shows TA survival (Fig. 3c).

To make it clearer, we have added explanations highlighted in green in the revised manuscript on page 9, changing ‘*Figure 5a shows temperature-dependent k_{lat} of α -MgAgSb and it demonstrates overall agreement with our experimental results of MgAg_{0.97}Sb_{0.99}^{15,38}*’ to ‘*Figure 5a shows the temperature-dependent k_{lat} of α -MgAgSb demonstrating the overall agreement between the calculation results of MgAgSb and the experimental results of MgAg_{0.97}Sb_{0.99}^{15,38}*’ and ‘*These modes could make the three-phonon process easier to occur (Fig. S7) and accordingly enhance the phonon anharmonicity of γ -MgAgSb*’ to ‘*The three-phonon process is easier to occur in γ -MgAgSb compared to α -MgAgSb demonstrated by the calculated three-phonon scattering phase space shown in Fig. S7, and accordingly enhance the phonon anharmonicity of γ -MgAgSb*’. The details of the demonstration of three-phonon processes and related references have been added in Supplementary Information, S9, in the revised manuscript.

We do hope that these answers are satisfactory. The revised manuscript is submitted separately. All changes in the revised manuscript are highlighted in green to distinguish the last revised version highlighted in yellow.

Sincerely yours,

Fangwei Wang

Baotian Wang

Huaizhou Zhao

Zhifeng Ren

REVIEWERS' COMMENTS:

Reviewer #3 (Remarks to the Author):

I have no further comments except, every figures and tables in supplementary should be cited sequently in a main manuscript. Figure S7 is, especially, not cited in the main manuscript.

Reviewer #3 (Remarks to the Author):

I have no further comments except, every figures and tables in supplementary should be cited sequently in a main manuscript. Figure S7 is, especially, not cited in the main manuscript.

Response: We thank Reviewer #3 for his/her third careful review of our work.

These figures of Figure S6, Figure S7, and Figure S8 are cited sequently in the revised manuscript.